# Designing minimal genomes using whole-cell models

Joshua Rees-Garbutt[1,2,7], Oliver Chalkley[1,3,4,7], Sophie Landon[1,3], Oliver Purcell[5], Lucia Marucci[1,3,6,8✉] & Claire Grierson[1,2,8✉]

In the future, entire genomes tailored to specific functions and environments could be designed using computational tools. However, computational tools for genome design are currently scarce. Here we present algorithms that enable the use of design-simulate-test cycles for genome design, using genome minimisation as a proof-of-concept. Minimal genomes are ideal for this purpose as they have a simple functional assay whether the cell replicates or not. We used the first (and currently only published) whole-cell model for the bacterium *Mycoplasma genitalium*. Our computational design-simulate-test cycles discovered novel in silico minimal genomes which, if biologically correct, predict in vivo genomes smaller than *JCVI-Syn3.0*; a bacterium with, currently, the smallest genome that can be grown in pure culture. In the process, we identified 10 low essential genes and produced evidence for at least two *Mycoplasma genitalium* in silico minimal genomes. This work brings combined computational and laboratory genome engineering a step closer.

[1] BrisSynBio, University of Bristol, Bristol BS8 1TQ, UK. [2] School of Biological Sciences, University of Bristol, Bristol Life Sciences Building, 24 Tyndall Avenue, Bristol BS8 1TQ, UK. [3] Department of Engineering Mathematics, University of Bristol, Bristol BS8 1UB, UK. [4] Bristol Centre for Complexity Science, Department of Engineering Mathematics, University of Bristol, Bristol BS8 1UB, UK. [5] Engine Biosciences, MBC Biolabs, 733 Industrial Road, San Carlos, CA 94070, USA. [6] School of Cellular and Molecular Medicine, University of Bristol, Bristol BS8 1UB, UK.. [7]These authors contributed equally: Joshua Rees-Garbutt, Oliver Chalkley [8]These authors jointly supervised this work: Lucia Marucci, Claire Grierson ✉email: lucia.marucci@bristol.ac.uk; claire.grierson@bristol.ac.uk

For genome engineering and design, minimal genomes are currently the best proof-of-concept[1]. These are reduced genomes containing only genes essential for life, provided there is a rich growth medium and no external stressors[1,2]. The greatest progress to date includes: JCVI-Syn3.0, a 50% gene reduction of Mycoplasma mycoides[2]; several strains of Escherichia coli reduced by 38.9[3] and 35%[4] of their base pairs in vivo; an E. coli gene reduction of 77.6% in Saccharomyces cerevisiae[5]; and two 36% gene reductions of Bacillus subtilis[6]. These efforts began with prescriptive design, gene selection using existing knowledge, or based on laboratory testing of individual genes, followed by iterative development. However, this process is time-consuming and expensive due to the limitations of current techniques and unexpected cell death. This hinders progress as laboratories can only follow a small number of high-risk research avenues, with limited ability to backtrack[1].

Another approach, building novel organisms from the bottom-up, is currently infeasible in the majority of bacteria due to technological and economic constraints[7]. Megabase sized genomes can be constructed within yeast[5,8], but one of the most promising approaches, genome transplantation, has only been demonstrated in a subset of Mycoplasma species[9–11] and is mutagenic[10].

A further barrier to genome minimisation is the dynamic nature of gene essentiality. A simple definition of a "living" cell is if it can reproduce; an "essential" gene being indispensable for cell division. A "non-essential" gene can be removed without preventing division[1,12]. But a cell's need for specific genes/gene products is dependent on the external environment and on the genomic context[1] (the presence or absence of other genes/gene products in the genome), which changes with each gene deletion. Some essential genes can become dispensable with the removal of another gene (e.g. a toxic byproduct is no longer produced, so its processing is unnecessary), referred to as "protective essential" genes[1,13,14]. Likewise, some non-essential genes become essential when a functionally-equivalent gene is removed, leaving a single pathway to a metabolite (a "redundant essential" gene pair). In addition, gene products can perform together as a complex, with individually non-essential genes involved in producing an essential function[15]; with enough deletions the remaining genes become essential. The cellular death that occurs when redundant essential genes are removed together, or complexes are disrupted, is referred to as synthetic lethality[2,16,17]. A recent review[1] updates gene essentiality from a binary categorisation to a four category gradient, where genes have: no essentiality (dispensable in all contexts), low essentiality (dispensable in some contexts, i.e. redundant essential and complexes), high essentiality (indispensable in most contexts, i.e. protective essential), and complete essentiality (indispensable in all contexts). These broad labels describe an individual gene's essentiality in different genomic contexts (conditional essentiality), and are compatible with other labels that explain underlying interactions in greater detail.

To address these problems, we use existing computational models with novel genome design algorithms to investigate 10,000 s of gene knockout combinations in silico. Testing potential genome reductions for lethal interactions at scale should produce functional in silico genomes, which can be implemented in vivo with a lower risk of failure.

We used the Mycoplasma genitalium (M. genitalium) whole-cell model[18], which presently represents the smallest culturable, self-replicating, natural organism[19]. A single cell is simulated from random biologically feasible initial conditions until the cell divides or reaches a time limit. The model combines 28 cellular submodels, with parameters from >900 publications and >1900 experimental observations, resulting in 79% accuracy for

single-gene knockout essentiality[18]. It is the only existing model of a cell's individual molecules that includes the function of every known gene product (401 of the 525 M. genitalium genes) making it capable of modelling genes in their genomic context[18]. One hundred and twenty-four genes of unknown function are not modelled in silico, but as some unknown function genes have proposed fundamental functions[20], these are assumed to be essential in vivo (so are added in our in vivo predictions). Outside of single-gene knockout simulations, the model has been used to investigate discrepancies between in silico and real-world measurements[18,21], design synthetic genetic circuits in cellular context[22], and make predictions about retargeting existing antibiotics[23].

We produced two genome design algorithms (Minesweeper and the Guess/Add/Mate Algorithm (GAMA)) which use the M. genitalium whole-cell model to generate minimal genome designs. Using these computational tools we found functional in silico minimal genomes which, if biologically correct, produce in vivo predictions between 33 and 52 genes smaller than the most recent predictions for a reduced Mycoplasma genome of 413 genes[17]. These predicted genomes are ideal candidates for further in vivo testing.

## Results

**Genome design tools minesweeper and GAMA.** Minesweeper and GAMA conduct whole-cell model simulations in three step cycles: design (algorithms select possible gene deletions); simulate (the genome minus those deletions); test (analyse resulting cell). Simulations that produce dividing cells proceed to the next cycle, increasing the number of gene deletions and producing progressively smaller genomes. Minesweeper and GAMA have generated 4620 and 53,451 in silico genomes, respectively (see Data availability section), but for brevity only the smallest genomes are presented.

Minesweeper is a four-stage algorithm inspired by divide and conquer algorithms[24]. It initially deletes genes in groups but eventually deletes individual genes, and only deletes non-essential genes (determined by single-gene knockout simulations, see the "Initial input" section). Excluding essential genes reduces the search area, making it capable of producing minimal genome size reductions within two days. It uses between 8 and 359 CPUs depending on the stage (see Methods), with data storage handled by user-submitted information and simulation execution conducted manually.

GAMA is a biased genetic algorithm[25]. It conducts two stages (Guess and Add) of only non-essential gene deletions, before including essential genes in the third stage (Mate). GAMA produces deletions that vary by individual genes, requiring 100–1000 s of CPUs. It takes two months to generate minimal genome size reductions, using between 400 and 3000 CPUs depending on the stage. This size of in silico experiment requires more time than is allowed on available supercomputers so the genome design suite[26] was developed to implement GAMA (see Methods).

**Initial input.** To generate an initial input for Minesweeper and GAMA we simulated single-gene knockouts in an unmodified M. genitalium in silico genome (as previously reported[18,21], Supplementary Data 1). Of the 401 in silico modelled genes 42 are RNA-coding, which were not selected for knockout. Mutants in each of the 359 protein-coding genes were simulated individually (10 replicates each), with 152 genes classified as non-essential and 207 genes classified as essential (i.e. producing a dividing or non-dividing in silico cell, respectively). The majority of genes (58%)

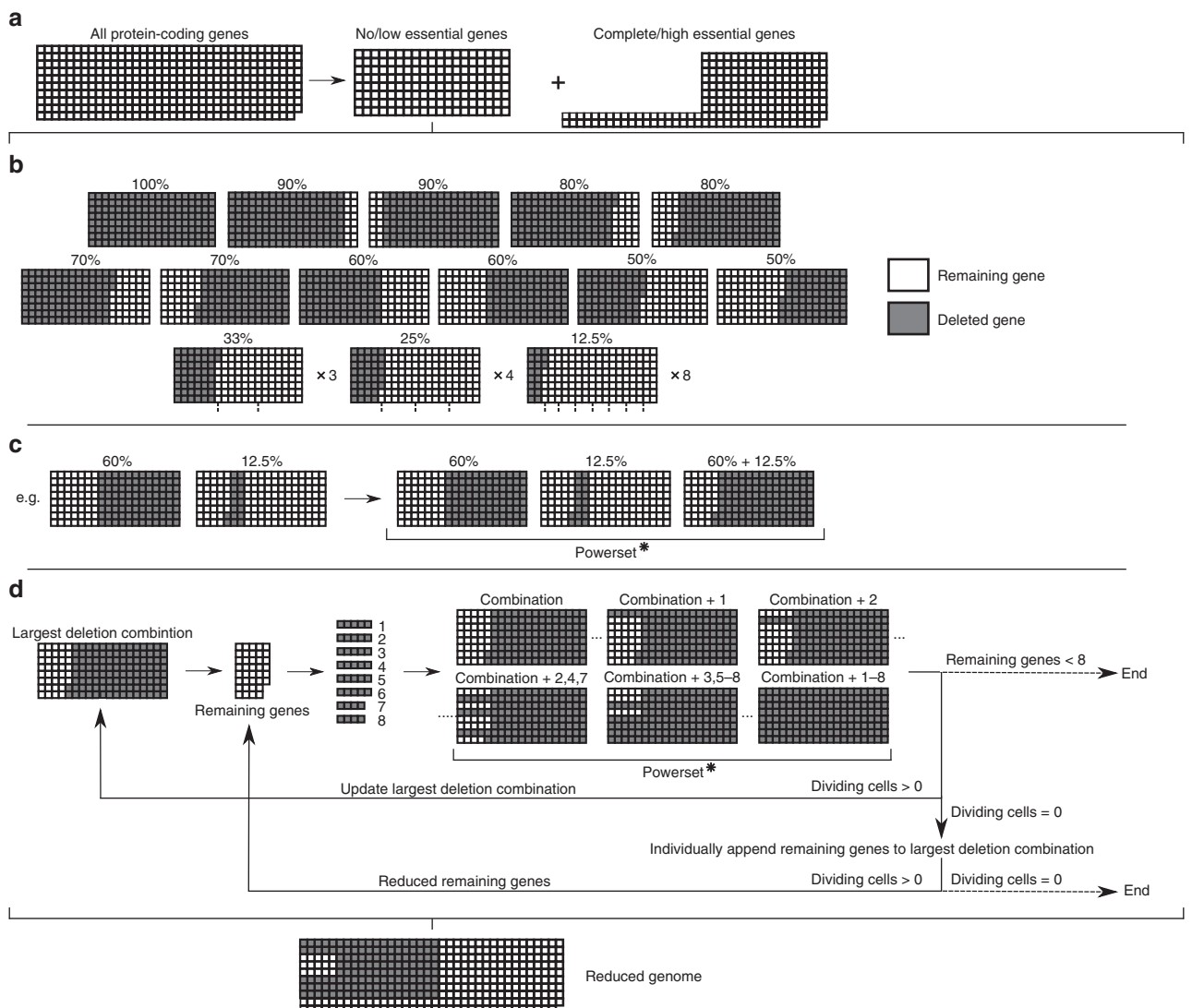

**Fig. 1 Minesweeper algorithm for genome design. a** In silico single-gene knockouts are conducted to identify no/low essential genes (whose knockout does not prevent cell division). **b** 26 deletion segments, ranging in size from 100 to 12.5% of the no/low essential genes, are simulated. Grey indicates a gene deletion, white indicates a remaining gene. Deletion segments that on removal do not prevent division go to the next stage. **c** The largest deletion segment is matched with all division-producing, non-overlapping segments. A powerset (all possible unique combinations of this set of matched deletion segments) is generated and each combination simulated. Combination segments that do not prevent division go to the next stage. **d** The largest combination segment determines the remaining no/low essential genes that have not been deleted. These remaining genes are divided into eight groups (see Methods), a powerset generated for these eight groups, and each member of the powerset individually appended to the current largest deletion combination and simulated. If none of these simulations on removal produces a dividing cell, the remaining genes are appended as single knockouts to the current largest deletion combination, removed and simulated. The individual remaining genes that do not produce a dividing cell are temporarily excluded and a reduced remaining gene list produced. Details of simulations settings are available in the Methods section. Simulation data generated by the Minesweeper algorithm is available[37] (see Data availability section). Powerset* = the complete powerset is not displayed here.

are essential; this was expected, as *Mycoplasma genitalium* is an obligate parasite with reduced genetic redundancy[27].

Three hundred and eighteen genes showed consistent phenotype across replicates, with 41 showing inconsistent phenotypes. Statistical analysis (binomial proportion confidence interval, Pearson–Klopper, 95% CIs for: a 6/10 replicate [5.74, 6.87], 7/10 replicates [6.66, 7.93], 8/10 replicates [7.56, 8.97], 9/10 replicates [8.45, 9.99]) resulted in assigning the most common phenotype (see Methods and Supplementary Data 2). Overall, our results agree 97% with Karr et al.[18] (Supplementary Data 3).

**Minesweeper algorithm and results**. The first stage of Minesweeper (Fig. 1) conducts individual protein-coding gene

knockouts (see Initial input section), removing complete/high essential genes as deletion candidates.

The second stage sorts the singly non-essential genes into deletion segments (12.5–100% of the remaining genes, resulting in 26 segments). Deletion segments that can be removed and still produce a dividing cell are carried forward.

The third stage progresses with the largest deletion segment that can be removed and produce a dividing cell, which is matched with other division-producing, non-overlapping deletion segments. A powerset is generated (i.e. a set containing all possible unique combinations of the matched deletion segments, including zero and individual deletion segments) and each of the deletion combinations is removed from an individual in silico cell and simulated.

The fourth stage is cyclical. The largest deletion combination from the third stage generates a remaining gene list (those yet to be deleted). The remaining genes are split into eight groups (see Methods) and a powerset is generated. Each deletion combination from the powerset is individually appended to the current largest deletion combination and simulated. The simulation results update the largest deletion combination, which is used to generate a new remaining gene list, starting the next cycle. If none of the deletion combinations produce a dividing cell, the remaining genes are singly appended to the largest deletion combination, removed and simulated. The individual remaining genes that do not produce a dividing cell are excluded for a cycle and a reduced remaining gene list is generated, which is used for the next cycle.

The fourth stage continues until there are eight or fewer remaining genes (where a final appended powerset is run) or all individually appended remaining genes do not produce a dividing cell. Both outcomes result in a list of deleted genes and identified low essential genes.

Minesweeper produced results quickly; within two days the third stage removed 123 genes (a 34% reduction), comparable to current lab-based efforts in other species[3,4,6]. In total, Minesweeper deleted 145 genes, creating an in silico *M. genitalium* cell containing 256 genes (named Minesweeper_256) and predicting an in vivo minimal genome of 380 genes. The in silico cell replicates DNA, produces RNA and protein, grows, and divides.

**GAMA algorithm and results**. The first and second stages of GAMA (Guess and Add) are pre-processing stages for the third stage (Mate), a genetic algorithm. Typically, a genetic algorithm would start with random gene knockouts, but the pre-processing stages produce large gene knockouts by exploiting patterns in the solution-space (Supplementary Data 14), i.e. any set of gene knockouts is significantly more likely to not produce division if it knocks out one or more singularly essential genes. Starting with large knockouts decreases the number of generations to produce minimal genome size reductions (Fig. 2).

In the first stage, Guess, all the non-essential genes from the initial input section are segmented into four sets (~40 genes). Each set is then used to generate ~400 subsets, by randomly choosing combinations of genes to delete that amount to 50–100% of the genes within the set. These are removed and simulated. If a cell divides, the deletion subset is labelled "viable" and carried forward.

During the second stage, Add, "viable" deletion subsets are randomly selected from two, three or four of the sets. These are combined into a larger deletion subset. Being able to select a varied number of sets decreases the chance of only producing non-dividing cells. Approximately three thousand combined subsets are created, removed and simulated. Those producing a dividing cell are ranked. The 50 smallest genomes (i.e. largest number of deletions) are carried forward.

The third stage, Mate, is cyclical. Each simulation "mates" two of the 50 smallest in silico genomes at random, and introduces random gene knockouts and knock-ins from a pool of all protein-coding genes (including complete and high essential genes). Each generation conducts 1000 simulations, which are ranked. The smallest 50 genomes is updated and passed to the next generation. The Mate step automatically stops after 100 generations, but was manually stopped at 46 generations after 20 generations without producing a smaller genome.

In total, the smallest GAMA-reduced in silico genome deleted 165 genes, creating an in silico *M. genitalium* genome of 236 genes (named GAMA_236), and predicting an in vivo minimal genome of 360 genes. GAMA removed more

genes than Minesweeper, while still producing a simulated cell which replicates DNA, produces RNA and protein, grows, and divides.

**Minesweeper_256, GAMA_236 and GAMA_237 genomes**. We investigated our two minimal genomes for consistency in producing a dividing in silico cell, and the range of behaviour they displayed. We simulated 100 replicates each of an unmodified *M. genitalium* in silico genome, Minesweeper_256, GAMA_236, and a single-gene knockout of a known essential gene (MG_006) to provide a comparison (Supplementary Data 7). The rate of division (or lack of in the MG_006 knockout simulations) was analysed to assign a phenotype penetrance percentage (quantifying how often an expected phenotype occurred). The unmodified *M. genitalium* and MG_006 knockout in silico genomes demonstrated consistent phenotypes (99% and 0% divided, respectively). Minesweeper_256 was slightly less consistent (89% divided), while GAMA_236 was substantially less consistent, producing a dividing in silico cell 18% of the time. This is not entirely unexpected given the presence of gene deletions that have high essentiality (see below, Supplementary Fig. 1).

We attempted to improve the division rate of GAMA_236 by conducting independent single knock-ins of its unique deletions (Supplementary Data 16). We found the highest division rate to be 33% (100 replicates) due to the reintroduction of a single gene (MG_270), creating the in silico minimal genome GAMA_237.

MG_270 (lipoate-protein ligase A) modifies MG_272 (Supplementary Note mmc1, pg. 65)[18], one of four genes (MG_271–274) that form the pyruvate dehydrogenase complex. This provides acetyl-CoA for the Krebs cycle, producing ATP. Reintroducing MG_270 repairs this complex and increases the available energy in the in silico cell.

While exploring further deletions for Minesweeper_256, another individual gene was found that impacted division rate. MG_104 (ribonuclease R), when deleted additionally, decreased the division rate to 1/9 of its original value (Supplementary Data 5). *Mycoplasma genitalium* has a very small pool of RNAs (Supplementary Note mmc1, pg. 85)[18], relying on ribonucleases to recycle RNAs for other cellular processes. Without ribonuclease R (the only modelled ribonuclease), the RNA decay submodel cannot function in silico, decreasing the amount of available RNAs.

The 100 replicates for each of the unmodified *M. genitalium* genome, Minesweeper_256, and GAMA_237 were plotted to assess the range of behaviour (Fig. 3). The unmodified *M. genitalium* whole-cell model (Fig. 3, top row) shows the range of expected behaviour for a dividing cell (in line with previous results[18]). Growth, protein production, and cellular mass increase over time, with the majority of cells dividing within 10 h (see cell diameter change). RNA production fluctuates but increases over time. DNA replication follows a characteristic shape, with some simulations delaying the initiation of DNA replication past ~9 h.

By comparison, Minesweeper_256 (Fig. 3, middle row) displays slower, and in some cases decreasing, growth over time which is capped at a lower maximum. Protein and cellular mass are generated more slowly, lower amounts are produced, and some erratic behaviour is present. The range of RNA production is narrower compared with the unmodified *M. genitalium* whole-cell model. DNA replication takes longer and initiation can occur later (at 11 h). Cell division occurs later, between 8 and 13.9 h. A number of simulations can be seen failing to replicate DNA and divide.

Compared with the other genomes, GAMA_237 (Fig. 3, bottom row) shows a much greater range of growth rates. Some

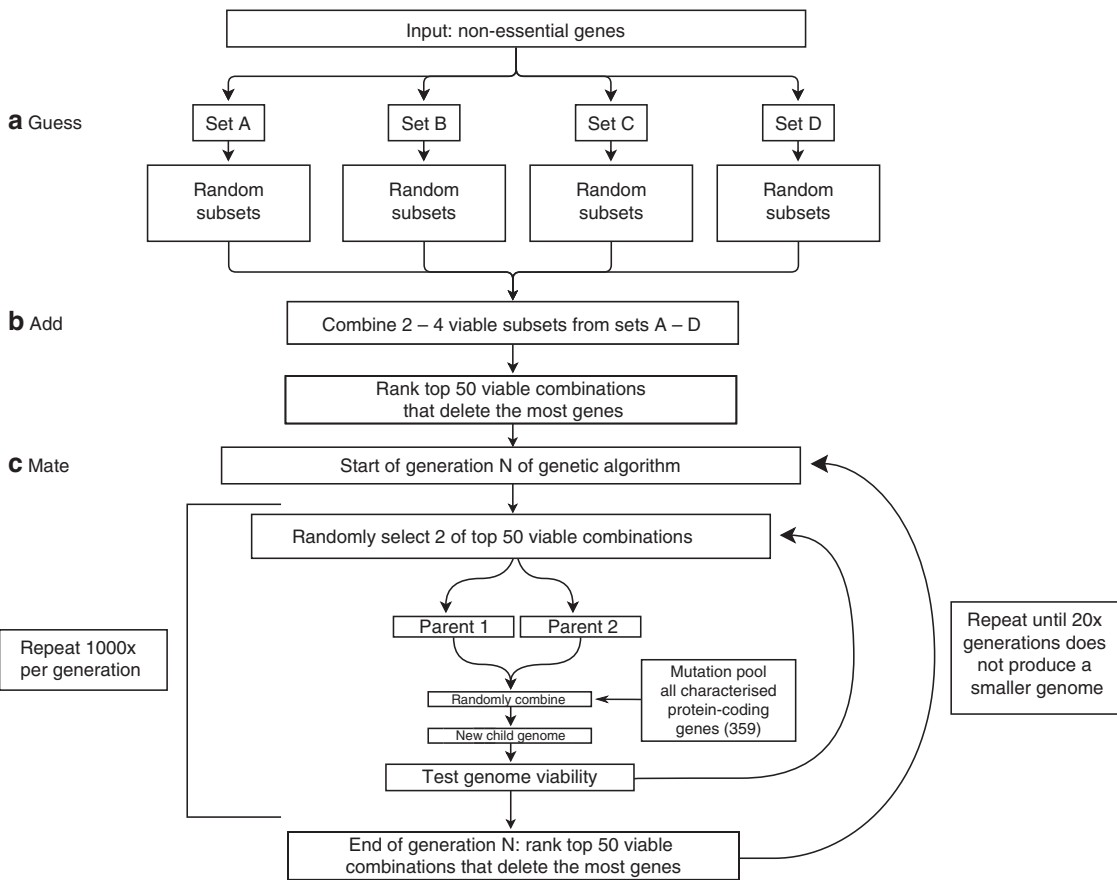

**Fig. 2 GAMA algorithm for genome design. a** Only non-essential genes whose knockout does not prevent cell division are deletion candidates and are equally divided into Sets A–D. Four hundred random deletion subsets are produced and simulated per set, each containing 50–100% of the genes within the set. Deletion subsets that do not prevent division ("viable") go to the next stage. **b** 3000 combinations of deletion subsets are generated and simulated. **c** This is a cyclical step. The mutation pool targets a random number of genes for alteration (both knock-ins and knockouts), including essential genes. Simulation data generated by the GAMA algorithm is available[37] (see Data availability section). Details of simulations settings are available in the Methods section.

grow as fast as the unmodified genome, some are comparable to Minesweeper_256, and some show very low or decreasing growth (this can also be seen in cellular mass). Observable protein levels appear between 2 and 5 h, followed by a slower rate of protein production in some simulations. The range of RNA production is reduced and the rate of RNA production is slower. Some simulations replicate DNA at a rate comparable to the unmodified genome, others replicate more slowly, and some do not complete DNA replication. Cell division occurs across a greater range of time (6–13.9 h). A number of simulations showing metabolic defects can be seen. These do not produce any growth and can be seen failing to divide.

**Genome analysis using gene ontology terms**. We investigated what processes were removed in the creation of Minesweeper_256 using gene ontology (GO) biological process terms (see Methods and Supplementary Data 9–11), standardised labels that describe a gene's function. The baseline *M. genitalium* whole-cell model has 259 genes of 401 genes (72% coverage) with GO terms on UniProt[28].

Minesweeper_256 has 186 (73%) genes with GO terms and 70 genes without. The 145 gene deletions reduced 22 (14%) GO categories, and removed 42 (27%) GO categories entirely, of which 30 were associated with a single gene (Supplementary Data 12).

The GO categories reduced include: DNA (repair, replication, topological change, transcription regulation and initiation);

protein (folding and transport); RNA processing; creation of lipids; cell cycle; and cell division. As the in silico cells continue to function, we can assume that these categories could withstand low-level disruption.

Removed GO categories that involved multiple genes include: proton transport; host interaction; DNA recombination; protein secretion and targeting to membrane; and response to oxidative stress. Removed GO categories that contain single genes include: transport (carbohydrate, phosphate and protein import, protein insertion into membrane); protein modification (refolding, repair, targeting); chromosome (segregation, separation); biosynthesis (coenzyme A, dTMP, dTTP, lipoprotein); breakdown (deoxyribonucleotide, deoxyribose, mRNA, protein); regulation (phosphate, carbohydrate, and carboxylic acid metabolic processes, cellular phosphate ion homeostasis); cell–cell adhesion; foreign DNA cleavage; SOS response; sister chromatid cohesion; and uracil salvage.

We conducted further analysis, as some of these removals could be of concern to the longevity of our in silico cell. The GO term proton transport applies to the genes MG_398–405, which form ATP synthase, an enzyme that generates ATP using energy from protons transferring across the cell membrane. This removes one pathway for producing ATP, but the minimal genome still contains intact phosphoglycerate kinase (MG_300) and pyruvate kinase (MG_216) that both produce ATP as part of glycolysis. In addition, there are 13 reversible reactions that produce ATP in the reverse reaction (Supplementary Data 17).

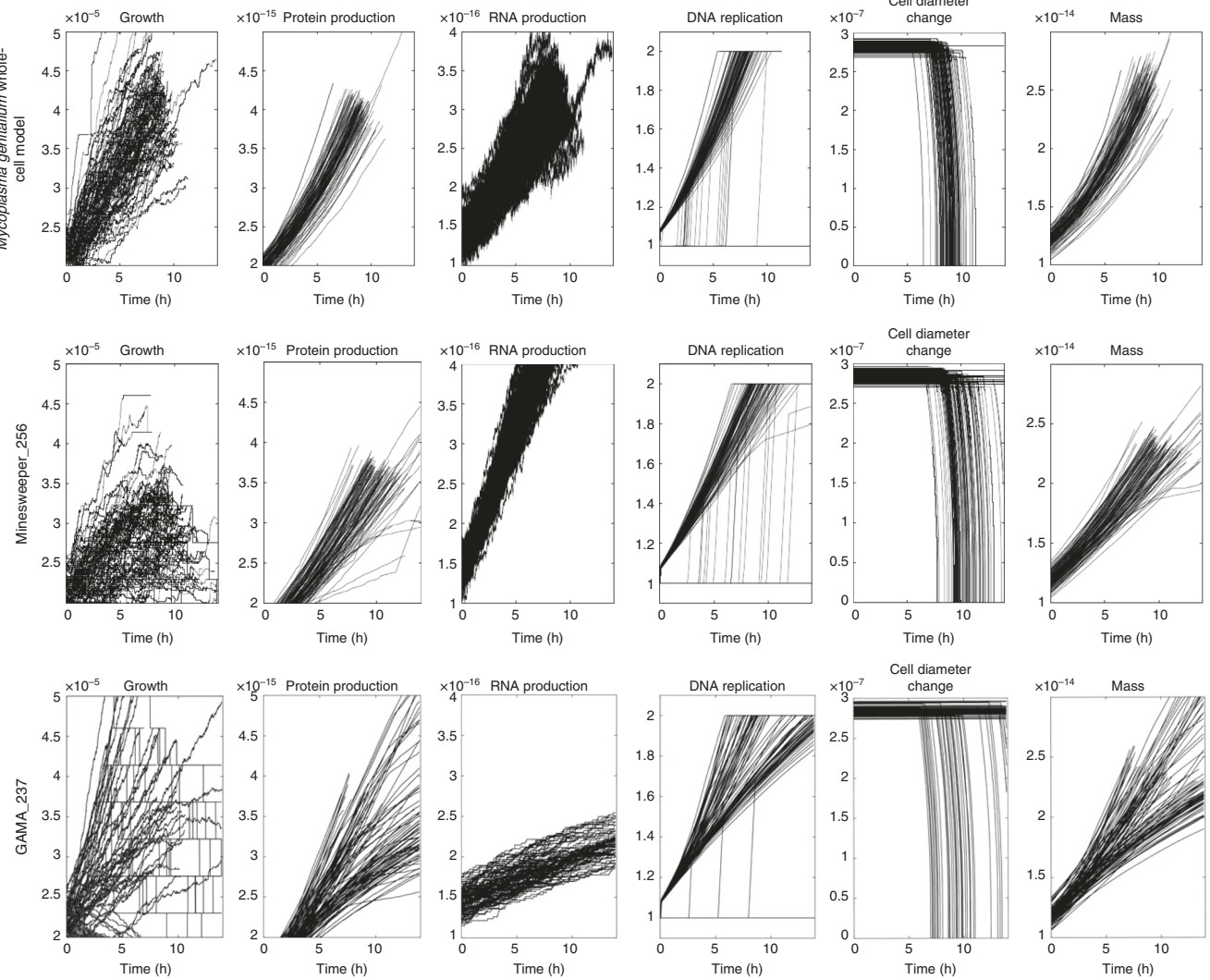

**Fig. 3 Behavioural comparison of whole-cell model, Minesweeper_256, and GAMA_237.** One hundred in silico replicates, with second-by-second values plotted for six cellular variables over 13.89 h (the default endtime of the simulations). The top row shows the expected cellular behaviour (previously show by Karr et al.[18]) and is used for comparison. Minesweeper_256 and GAMA_237 show deviations in phenotype caused by gene deletions. Non-aggregated data for each in silico simulation is available[37].

The GO term DNA recombination applies to the genes MG_339, MG_352, MG_358, MG_359, which conduct half of the steps in homologous recombination double strand break repair. This process, as well as nucleotide excision repair and base excision repair are removed from the cell. However, direct damage reversal (MG_254) and DNA polymerase (MG_001) are still present. *DisA* (MG_105), the DNA damage sensor, has currently been deleted but we believe this is due its model implementation (Algorithm S6 infers it is not a requirement for successful repair (Supplementary Note mmc1, pg. 43)[18]. This may be an inaccuracy in the model, and with its reintroduction the minimal genome would contain a functioning DNA repair process.

The GO term chromosome segregation applies only to MG_213, where it is listed as its tertiary function. The genes in the model that actually conduct chromosome segregation (MG_470, MG_221, MG_387, MG_384, MG_203, MG_204, MG_224 (Supplementary Note mmc1, pg. 34)[18] are all present in the minimal genome, but do not have an associated GO term. This underlines the use of caution when using GO terms and the need for secondary analysis.

The gene deletions in Minesweeper_256 reduce the ability of the in silico cell to interact with the environment and defend against external forces. They also cause a reduction in control, from transport to regulation to genome management, and prune metabolic processes and metabolites. This leaves the in silico cell alive, but more vulnerable to external and internal pressures, less capable of responding to change, and more reliant on internal processes occurring by chance.

In comparison, GAMA_237 has 163 genes (69% coverage) with GO terms on UniProt[28], with 73 genes with no GO terms. The 165 genes deleted reduced 18 (11%) GO categories, and removed 54 (35%) GO categories, 38 of which were associated with a single gene (Supplementary Data 13). The gene deletions unique to GAMA_237 can be seen in Table 1 and Table 2.

One reduced GO category was less affected compared with Minesweeper_256 (glycerol metabolic process) and one unaffected GO category was unique to GAMA_237 (phosphate ion transmembrane transport). Three GO categories were reduced further in GAMA_237: DNA transcription, DNA transcription regulation, and transport (ABC transporters, see Table 1).

**Table 1 Low essential genes from Minesweeper_256 and GAMA_237 genomic contexts.**

| Gene | Annotation | Function | Removed in | Present in |
|------|-----------|----------|-----------|-----------|
| MG_039 | Uncharacterised | Probable catalyst of redox reactions. | GAMA_237 | Minesweeper_256 |
| MG_289 | p37 | High-affinity transport system protein attached to cell membrane. | GAMA_237 | Minesweeper_256 |
| MG_290 | p29 | Probable ATP-binding cassette (ABC) transporter. | GAMA_237 | Minesweeper_256 |
| MG_291 | p69 | Permease (ABC membrane transporter) protein. | GAMA_237 | Minesweeper_256 |
| MG_427 | Unnamed | Reduces peroxides, protecting against oxidative stress. | GAMA_237 | Minesweeper_256 |
| MG_033 | glpF | Facilitates glycerol across the membrane. | Minesweeper_256 | GAMA_237 |
| MG_410 | pstB | Imports phosphate (part of PstSACB ABC complex). | Minesweeper_256 | GAMA_237 |
| MG_411 | pstA | Permease protein for phosphate transport system. | Minesweeper_256 | GAMA_237 |
| MG_412 | Uncharacterised | Probable phosphate ion binding attached to cell membrane. | Minesweeper_256 | GAMA_237 |
| MG_305 | dnaK | Chaperone protein involved in refolding mis/unfolded heat shock proteins. | M.g* whole-cell model | GAMA_237 and Minesweeper_256 |

Protein annotation and function obtained from UniProt[28], based on Fraser et al.'s *Mycoplasma genitalium*\* G37 genome[19].

**Table 2 High essential genes from GAMA_237 genomic contexts.**

| Gene | Annotation | Function |
|------|-----------|----------|
| Transcription-related | | |
| MG_022 | rpoE | DNA-directed RNA polymerase subunit delta. Presence causes increased specificity of transcription, a decreased affinity for nucleic acids, and enhanced recycling. |
| MG_141 | nusA | Transcription termination/antitermination protein. Participates in both. |
| MG_177 | rpoA | DNA-directed RNA polymerase subunit alpha. Catalyzes the transcription of DNA into RNA using the four ribonucleoside triphosphates as substrates. |
| MG_249 | sigA | RNA polymerase sigma factor. The primary initiation factor during exponential growth, promoting the attachment of RNA polymerase to specific sites. |
| MG_282 | greA | Transcription elongation factor. Cleaves the fraction of nascent transcripts that get trapped at arresting sites, resuming elongation and allowing efficient RNA polymerase transcription. |
| MG_340 | rpoC | DNA-directed RNA polymerase subunit beta. Catalyzes the transcription of DNA into RNA using the four ribonucleoside triphosphates as substrates. |
| MG_341 | rpoB | Additional part of DNA-directed RNA polymerase subunit beta. |
| Translation-related | | |
| MG_008 | mnmE | tRNA modification GTPase. Addition of a carboxymethylaminomethyl group to certain tRNAs. |
| MG_084 | tilS | tRNA(Ile)-lysidine synthase. Ligates lysine to the AUA codon-specific tRNA, changing the amino acid specificity from methionine to isoleucine. |
| MG_182 | truA | tRNA pseudouridine synthase A. Forms pseudouridine in the anticodon stem and loop of tRNAs. |
| MG_295 | mnmA | tRNA-specific 2-thiouridylase. Catalyzes 2-thiolation of uridine in tRNAs. |
| MG_347 | trmB | tRNA methyltransferase. Catalyzes the formation of N7-methylguanine in tRNA. |
| MG_367 | rnc | Ribonuclease 3. Produces ribosome large and small RNAs (23S and 16S). Processes some mRNAs and tRNAs. Digests double-stranded RNA. Other rRNA processing genes: MG_110, MG_139, MG_425 also removed in GAMA_237. |
| MG_372 | thiI | tRNA sulfurtransferase. Catalyzes the transfer of sulfur to tRNAs to produce 4-thiouridine, and catalyzes the transfer of sulfur to carrier protein ThiS (a step in the synthesis of thiazole). |
| MG_379 | mnmG | Forms a tetramer with MG_008. Addition of a carboxymethylaminomethyl group to certain tRNAs. |
| MG_445 | trmD | tRNA methyltransferase. Specifically methylates guanosine−37 in various tRNAs. |
| MG_465 | rnpA | Ribonuclease P protein component. Produces mature tRNAs (catalyzes removal of 5'-leader sequence). In addition, it broadens the substrate specificity of the ribozyme through binding. |

Protein annotation and function obtained from UniProt[28], based on Fraser et al.'s *Mycoplasma genitalium*\* G37 genome[19].

Categories that were removed solely in GAMA_237 include: DNA transcription (termination, regulation, elongation, anti-termination, initiation); tRNA (processing, modification); and rRNA catabolic process.

The GO analysis of GAMA_237, when compared with that of Minesweeper_256, suggests a further reduction of both internal control and reactivity to external environment. These reductions are discussed further below.

**Low essential genes**. We analysed Minesweeper_256 and GAMA_237 to determine whether these were different minimal genomes or GAMA_237 was an extension of Minesweeper_256. We compared unmodified *M. genitalium*, Minesweeper_256, and GAMA_237 genomes (Fig. 4, Supplementary Data 6), which highlighted gene deletions unique to each minimal genome. We additionally compared Minesweeper_256 to all of the GAMA in silico genomes that deleted 145–165 genes (256–236 genes in the in silico genome). Figure 5 shows the GAMA algorithm's avenue of gene reductions converging to a minimal genome, with Minesweeper_256 not on the same minimisation path.

Our genome comparison found that Minesweeper_256 removed four genes, and GAMA_237 removed five genes (Table 1), that could not be removed from the other genome (either individually or as a group) without preventing cellular division (Supplementary Data 5). An additional gene, MG_305, could not be removed in either GAMA_237 or Minesweeper_256. We confirmed that these ten genes were individually non-essential (Supplementary Data 1), and that nine of the genes have low essentiality[1]. To identify the cause of this synthetic

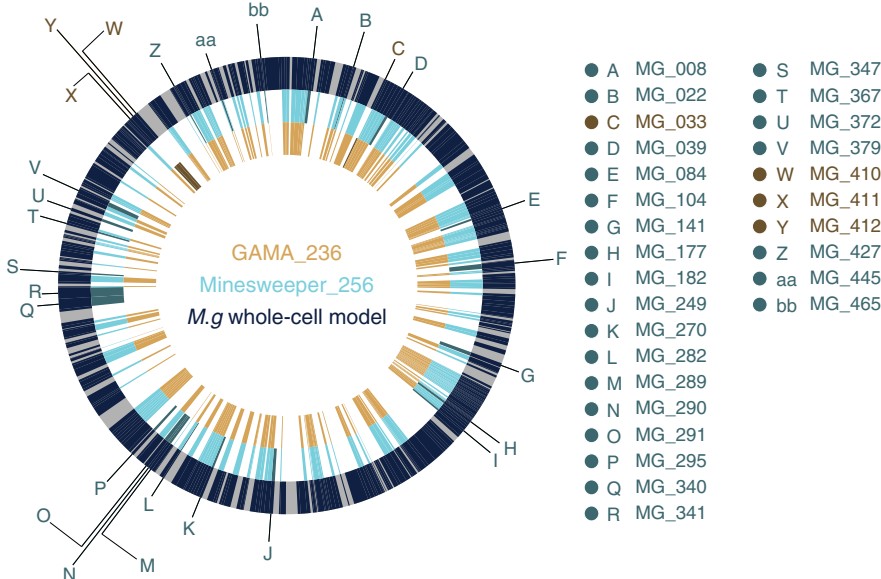

**Fig. 4 Genome comparison of whole-cell model, Minesweeper_256, and GAMA_237.** The outer ring displays the *M. genitalium* genome (525 genes in total), with modelled genes (401) in navy and unmodelled genes (124, with unknown function) in grey. The middle ring displays the reduced Minesweeper_256 (256 genes) genome in light blue, with genes present in Minesweeper_265 but not in GAMA_237 in dark blue. The inner ring displays the reduced GAMA_237 (237 genes) genome in light yellow, with genes present in GAMA_237 but not in Minesweeper_265 in dark yellow. Figure produced from published *M. genitalium* genetic data[18,19], with genetic data for Minesweeper_256 and GAMA_237 available in Supplementary Data 5.

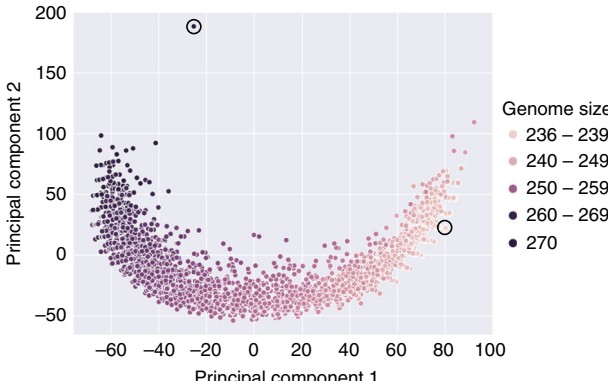

**Fig. 5 Comparing Minesweeper_256 and 2954 GAMA genomes.** The genomes of Minesweeper_256 and all the genomes found by GAMA that were 270 genes and smaller were collated[37]. Each point represents a single genome and is plotted based on a similarity metric (see Methods), showing the pathway of convergence to GAMA_237. The circled genome in the top left is Minesweeper_256 and the circled genome in the bottom right is GAMA_237. The binary formatted genome data used to produce this figure is available[37].

lethality we attempted to match the functions of the low essential genes together, anticipating redundant essential gene pairs or groups.

We found two genes in GAMA_237 (MG_289, MG_291) that had matching GO terms with the gene MG_411 in Minesweeper_256. These, and three other adjacent genes on the genome, were tested by combinatorial gene knockouts in an unmodified *M. genitalium* whole-cell model genome (Supplementary Data 8). MG_289, MG_290, MG_291 were found to form a functional group, as were MG_410, MG_411, MG_412. These genes could be deleted individually and in functional groups from an otherwise unmodified *M. genitalium* whole-cell genome, and produce a dividing in silico cell. However, any double gene

deletion combination that involved one gene from each functional group resulted in a cell that could not produce RNA, produce protein, replicate DNA, grow or divide.

*M. genitalium* only has two external sources of phosphate, inorganic phosphate and phosphonate. MG_410, MG_411, and MG_412 transport inorganic phosphate into the cell, and MG_289, MG_290, and MG_291 transport phosphonate into the cell (Supplementary Table 2[27]). These phosphate sources proved to be a key difference between our minimal genomes. Minesweeper_256 removed the phosphate transport genes, relying on phosphonate as the sole phosphate source. GAMA_237 removed the phosphonate transport genes, relying on inorganic phosphate as the sole phosphate source. This can be seen in the GO term analysis, the phosphate ion transmembrane transport is still present in GAMA_237 but not in Minesweeper_256 (Supplementary Data 12, 13).

It has previously been theorised that individual bacterial species will have multiple minimal genomes[29,30], with different gene content depending on the environment (it is predicted that an additional 500 genes would be required to survive on minimal media[31], see Methods for in silico conditions) and which evolutionarily redundant cellular pathways were selected during reduction. We propose that one of these selections is the sourcing of phosphate, with minimal genomes differing by choice of phosphate transport genes and associated processing stages, equivalent to the *phn* gene cluster in *Escherichia coli*[32]. We could not however find any annotated phosphonate processing genes that had been subsequently removed in GAMA_237. We suspect that further "pivot points" (the selection of one redundant cellular pathway over another during reduction) will be identified in future in vivo and in silico bacterial reductions, increasing the number of minimal genomes per bacterial species.

We additionally investigated MG_305, the gene that neither Minesweeper_256 or GAMA_237 could remove (Table 1, Supplementary Data 15). Four other genes share the protein folding GO term, MG_019, MG_201, MG_238, and MG_393. Three were unmodified in either genome, but MG_393 was

removed by both Minesweeper_256 and GAMA_237, indicating a potential redundant essential relationship. However, knocking in MG_393 and knocking out MG_305 produces a non-dividing cell, in both GAMA and Minesweeper. We theorise that MG_305 has additional redundant pair relationships that have already suffered one deletion shared by both GAMA_237 and Minesweeper_256 (which share 141 gene deletions).

**High essential genes**. Our comparison of the genomes also found 17 genes knocked out in GAMA_237 that have high essentiality[1] (Table 2). They were defined as essential by single knockout in an unmodified *M. genitalium* whole-cell model, but could be removed in the genomic context of GAMA_237 without preventing division (Supplementary Data 1, 5). We also found that four of these 17 genes could be removed as a group in the genomic context of Minesweeper_256, but doing so greatly increased the number of non-dividing cells produced (Supplementary Data 5 (col D,S,T)).

These 17 genes can be grouped into either transcription-related or translation-related functions. The transcription-related genes produce enzymes required for transcription in the model (Supplementary Note mmc1, pg. 93[18]), all of which have been removed in the GAMA_237 minimal genome. In addition, we found that the five modelled transcriptional regulators (Supplementary Table S3P col D[18]) were removed from GAMA_237. This removes the process of transcription from the in silico cell.

The translation-related genes are involved in the two parts of the core translation machinery (consisting of ribosome synthesis, tRNA maturation, and tRNA aminoacylation[33]), with only tRNA aminoacylation being conserved in GAMA_237 (Supplementary Note mmc1, pg. 104[18]). Nine genes involved in tRNA maturation are removed (Supplementary Table S3AB[18]), and a key gene in ribosome synthesis (MG_367) is deleted (Supplementary Note mmc1, pg. 89[18]) from GAMA_237. This effectively removes the process of translation from the in silico cell.

We theorise that the lower in silico division rate for GAMA_237 (33%) is due to the cell being reliant on the biologically feasible random initial conditions (i.e. already present RNAs and proteins) to survive a single generation. This underlines a problem with using in silico models in genome design. To the best of our knowledge, the *Mycoplasma genitalium* whole-cell is modelled correctly, including the implementation of deleting genes (Supplementary Note mmc1, pg. 117[18]; Simulation.m, lines 194–350[34]). However, with the complete model only capable of modelling a single generation, in silico cells that would not produce a dividing second generation in silico cell (or a functioning in vivo cell) can divide successfully and appear functional. Modelling multiple generations is required to allow the production of in silico genome designs that could reliably predict in vivo genome function. The *E. coli* whole-cell model[35] (code available, currently in review) models multiple generations, as should all future whole-cell models.

## Discussion
We created two genome design algorithms (Minesweeper and GAMA) that used computational design-simulate-test cycles to produce in silico *M. genitalium* minimal genomes (Minesweeper_256 and GAMA_237, 36% and 41% in silico reductions, respectively). If biologically correct, our subsequent in vivo minimal genome predictions are smaller than *JCVI-syn3.0* (currently the smallest genome that can be grown in pure culture at 473 genes[2]) and smaller than the most recent predictions for a reduced *Mycoplasma* genome (413 genes)[17]. In addition, we identified 10 low essential genes[1], and produced evidence for at least two minima for *M. genitalium* in silico.

We believe that single-gene knockout classifications are unreliable for genome minimisation, as they fail to take into account genomic context. Single-gene knockout studies will incorrectly estimate minimal genome size, as low essential genes will be scored as non-essential[2,16,17] and if high essential genes are present they will be scored as essential.

There are limitations to the approach presented here. Models are not perfect representations of reality. Through necessity, this model bases some of its parameters on data from other bacteria[18], complete multi-generation simulations are not possible, and *M. genitalium* has genes of unknown function that the model cannot account for. The success of our in silico genomes in vivo is dependent on the accuracy of the model, which is untested at this large scale of genetic modification.

Further specific issues have been highlighted, including the modelling of *DisA*, and the biologically-infeasible removal of the high essential genes in silico (an outcome of the model's single generation lifespan). We do not have confidence in the in silico high essential or *DisA* gene deletions. However, the gene deletions shared by GAMA_237 and Minesweeper_256 (141 gene deletions) and the deletions responsible for each of the phosphate "pivot points" are worthy of in vivo testing.

We attempted to gain further insight by using BLAST (tblastn) to compare the shared deletions to *JCVI-Syn3.0* (Supplementary Data 18). We matched 56% of *JCVI-Syn3.0* genes to the *Mycoplasma genitalium* whole-cell modelled genes, finding that 73 of the 141 shared deletions had no BLAST match with *JCVI-Syn3.0*. There were 15 deletions in common and 53 deletions not removed in *JCVI-Syn3.0*. We conclude that an explicit comparison between the reduced *Mycoplasma* genomes is difficult and inconclusive, mainly due to the differences in the considered species (*JCVI-Syn3.0* is a reduction of *JCVI-Syn1.0*, which is based on *Mycoplasma mycoides*).

We believe that our in silico shared deletions could predict a viable *Mycoplasma genitalium* minimal cell. Given that the impact of the unmodelled genes is unknown (e.g. if they perform a unique essential function with a gene/gene product that has already been removed the in vivo cell will not survive), until these predictions are tested experimentally we cannot firmly state how long our predicted reduced in vivo cells would survive and replicate, and whether they represent a truly minimal *M. genitalium* genome.

Our algorithms are currently adaptable to in review and future whole-cell models, as the algorithms interact with the models only via the input of gene deletion lists and analysing the output. With future, multi-generational, whole-cell models we will have greater confidence that our algorithms have produced in silico genome designs that will be viable in vivo. This includes the *E. coli* whole-cell model at the Covert Lab, Stanford[35] and the *Mycoplasma pneumoniae* whole-cell model at the Karr Lab, Mount Sinai, New York[36].

We believe that a hybrid of computational and lab-based genome design and construction is now in sight. This could produce quicker and cheaper laboratory results than currently possible, opening up this research to broader and inter-disciplinary research communities. It also expands our research horizons raising the possibility of building truly designer cells, with increased efficiency and functional understanding.

## Methods
**Model availability**. The *M. genitalium* whole-cell model is freely available: https://github.com/CovertLab/WholeCell. The model requires a single CPU and can be run with 8 GB of RAM. We run the *M. genitalium* whole-cell model on Bristol's supercomputers using MATLAB R2013b, with the model's standard settings.

However, we use our own version of the SimulationRunner.m. MGGRunner.m is designed for use with supercomputers that start hundreds of simulations simultaneously, artificially incrementing the time-date value for each simulation, as

this value is subsequently used to create the initial conditions of the simulation. This incrementation prevents the running of multiple simulations with identical initial conditions.

Our research copy of the whole-cell model was downloaded 10th January 2017.

**Statistics**. We used the R binom package (https://www.rdocumentation.org/packages/binom) to conduct one-tailed binomial proportion confidence intervals on our 41 genes showing inconsistent results (success ranging from 6 to 9 replicates, out of a total of 10 replicates). We used binom.confit.exact (Pearson–Klopper) using 95% CIs, producing 6/10 replicates [0.26, 0.87], 7/10 replicates [0.34, 0.93], 8/10 replicates [0.44, 0.97], 9/10 replicates [0.55, 0.99]). We graphed these results in R and in Python using Seaborn (https://seaborn.pydata.org/), the exact values, code, and graphs produced are available in Supplementary Data 2.

Figure 5 was generated by creating a similarity matrix between all of the 2955 genomes, with the gene information represented in a binary format (present or absent)[37]. The matrix calculated a distance metric (1 – (the genes that are the same between genomes/total number of genes)), with each genome comparison given a normalised score (0 = the genomes were identical, 0.5 = as different as would be expected if each genome was generated randomly, 1 = completely different). The resulting $2955 \times 2955$ matrix was then reduced to two dimensions with a standard PCA.

**Mycoplasma genitalium in silico environmental conditions**. *Mycoplasma genitalium* is grown in vivo on SP4 media. The in silico media composition is based on the experimentally characterized composition, with additional essential molecules added (nucleobases, gases, polyamines, vitamins, and ions) in reported amounts to support in silico cellular growth. In addition, the *M. genitalium* whole-cell model represents 10 external stimuli including temperature, several types of radiation, and three stress conditions. For more information see Supplementary Tables S3F, S3H, S3R[18].

**Minesweeper**. Minesweeper is written in Python3 and consists of four scripts (one for each stage). It uses no external libraries, so should be able to be run on any modern operating system (as they come with Python preinstalled) via a terminal. Each stage/script requires a text file(s) as input, with each stage outputting simulation files. These are run on a supercomputer and the automatically produced summary file is used as input for the next stage. Stages one to three are sequential, with stage four repeating until Minesweeper stops, with progress recorded in the deletion log in /OUTPUT_final. More detailed information, including instructions for the demo, is provided in the README (https://github.com/GriersonMarucciLab). The test data provided with the demo was produced by completing stages one to four in Minesweeper (producing 2310 in silico minimal genomes with stage four repeating three times), using the averaged single gene essentiality data from Supplementary Data 1 (column F).

The first stage of Minesweeper is optional, i.e. if you already have single-gene knockout simulation results, you can proceed to the second stage. The second stage creates 26 deletion segments: 100%, 90%A, 90%B, 80%A, 80%B, 70%A, 70%B, 60% A, 60%B, 50%A, 50%B, 33%A–C, 25%A–D, 12.5%A–H. The A segments start from the top of the list of genes, whereas the B segments start from the bottom of the gene list. The third stage progresses with the three largest deletion segments that produced a dividing cell, these three variants are referred to as red, yellow, blue. These perform as replicates and as a check on if the results are converging. The three variants are matched with smaller, dividing, non-overlapping segments using a list of allowed matches (implementation is detailed in third stage script), and unique combinations generated using a Python implementation of powersets. The fourth stage splits the remaining genes into eight groups. The reason for selecting eight groups and three variants is that a set of eight produces 256 unique combinations. Three variants each with 256 simulations (768 total) represents 85% of the capacity of BlueGem. A set of nine groups with three variants (1536 simulations total) is 170% the capacity of BlueGem. Queueing systems mean that you do not require this number of CPUs in total, but the execution time is multiplied as you wait for the simulations to process. The number of variants and groups can be lowered or increased depending on the number of CPUs available. To do so, make changes to the calculations and list generation in the eightPanelGroupingsGeneration function in the fourth script.

**GAMA**. GAMA is written in Python3 and relies on a variety of different packages. These dependencies can be easily taken care of by installing it from PyPI using either 'pip install genome_design_suite' or 'conda install genome_design_suite' (it is recommended that you do this from within a virtual environment since this is pre-alpha and has not been extensively tested with different versions of all the libraries). A dependencies list is available in the main directory of the github repository (https://github.com/GriersonMarucciLab) if you would like to do this manually. The main dependency is the 'genome_design_suite'[26] which is a suite of tools that we created at the University of Bristol which enables it to be easily run on different (or even multiple) clusters and allows automatic data processing and database management. Due to the large amount of data produced by the whole-cell model, the simulation output data was reduced to essential data, converted into Pandas DataFrames (https://pandas.pydata.org/) and saved in Pickle files. GAMA

would have produced 100 s of TBs of data in the model's native output format (compressed Matlab files) which we are not able to store so this was an essential step. In order to run this code you must have a computer dedicated to remotely manage the simulations. A PC with a quad-core Intel(R) Xeon(R) CPU E5410 (2.33 GHz) and 1GB of RAM running CentOS−6.6 was used as our computer manager, which is referred to as OC2. GAMA was run on OC2 using the scripts contained in gama_management.zip. Each stage of GAMA was run individually and manually updated as it was in proof-of-concept stage when GAMA_236 was found. ko.db is an SQLite3 database used to stored key information about simulations like cell average growth rate and division time.

The guess stage splits the singularly non-essential genes in roughly equally sized partitions. The four files, focus_on_NE_split_[1–4].py, run the exploration of each of the four partitions of the guess stage from OC2, after unzipping gama_management.zip these can be found in gama/guess. The submission scripts and other files automatically created to run the simulations on the cluster can be found in gama_run_files.zip → gama_run_files/guess. The simulation output is saved in Pickle files and can be found in gama_data/guess. viability_of_ne_focus_sets_pickles.zip contains the viability data of these simulations and the Python script used to collect it.

The add stage was executed on OC2 by running the files in gama_management. zip → gama/add. The submission scripts and other files automatically created to run the simulations on the cluster can be found in gama_run_files.zip → gama_run_files/add. The simulation output can be found in gama_data/add and an overview of the simulation results can be found in ko.db where the batchDescription.name is some derivative of 'mix_ne_focus_split'.

The mate stage was executed on OC2 by running the file in gama_management. zip → gama/mate. The submission scripts and other files automatically created to run the simulations on the cluster can be found in gama_run_files.zip → gama_run_files/mate. The simulation output can be found in gama_data/mate and an overview of the simulation results can be found in ko.db where batchDescription.name is some derivative of 'big_mix_of_split_mixes'.

**Equipment**. We used the University of Bristol Advanced Computing Research Centres's BlueGem, a 900-core supercomputer, which uses the Slurm queuing system, to run whole-cell model simulations. GAMA also used BlueCrystal, a 3568-core supercomputer, which uses the PBS queuing system.

We used a standard office desktop computer, with 8 GB of RAM, to write new code, interact with the supercomputer, and run single whole-cell model simulations. We used the following GUI software on Windows/Linux Cent OS: Notepad++ for code editing, Putty (ssh software)/the terminal to access the supercomputer, and FileZilla (ftp software) to move files in bulk to and from the supercomputer. The command line software we used included VIM for code editing, and SSH, Rsync and Bash for communication and file transfer with the supercomputers.

**Data format**. The majority of output files are state-NNN.mat files, which are logs of the simulation split into 100-s segments. The data within a state-NNN.mat file are organised into 16 cell variables, each containing a number of sub-variables. These are typically arranged as 3-dimensional matrices or time series, which are flattened to conduct analysis. The other file types contain summaries of data spanning the simulation.

**Data analysis process**. The raw data are automatically processed as the simulation ends. runGraphs.m carries out the initial analysis, while compareGraphs.m overlays the output on collated graphs of 200 unmodified *M. genitalium* simulations. Both outputs are saved as MATLAB .fig and .pdfs, though the .fig files were the sole files analysed. The raw .mat files were stored in case further investigation was required.

To classify our data we chose to use the phenotype classification previously outlined by Karr (Fig. 6b[18]), which graphed five variables to determine the simulated cells' phenotype. However, the script responsible for producing Fig. 6b, SingleGeneDeletions.m, was not easy to modify. This led us to develop our own analysis script recreating the classification: runGraphs.m graphs growth, protein weight, RNA weight, DNA replication, cell division, and records several experimental details. There are seven possible phenotypes caused by knocking out genes in the simulation: (i) non-essential if producing a dividing cell, (ii) slow growing if producing a dividing cell slowly; and essential if producing a non-dividing cell because of a (iii) DNA replication mutation, (iv) RNA production mutation, (v) protein production mutation, (vi) metabolic mutation, or (vii) division mutation.

For the single-gene knockout simulations produced in initial input section, the non-essential simulations were automatically classified and the essential simulations flagged. Each simulation was investigated manually and given a phenotype using the decision tree (Supplementary Data 4).

For in silico experiments conducted using Minesweeper and GAMA, simulations were automatically classified solely by division, which can be analysed from cell width or the end time of the simulation.

Further analysis, including: cross-comparison of single-gene knockout simulations, comparison to Karr et al.'s[18] results, analysis of Minesweeper and

GAMA genomes (genetic content and similarity, behavioural analysis, phenotypic penetrance, gene ontology), and identification and investigation of high and low essentiality genes and groupings, were completed manually. The GO term analysis of gene deletion impacts was processed by a created script (https://github.com/GriersonMarucciLab), then organised into tables of GO terms that were classified as unaffected, reduced, or removed entirely.

**Modelling scripts, process and simulations**. Generally, there are six scripts we used to run the whole-cell model. Three are the experimental files created with each new experiment (the bash script, gene list, experiment list), and three are stored within the whole-cell model and are updated only upon improvement (MGGrunner.m, runGraphs.m and compareGraphs.m). The bash script is a list of commands to carry out. Each new bash script is created from the TemplateScript.sh template, which determines how many simulations to run, where to store the output, which analysis to run, and where to store the results of the analysis. The gene list is a text file containing rows of gene codes (in the format 'MG_XXX'). Each row corresponds to a single simulation and determines which genes that simulation should knockout.

The experiment list is a text file containing rows of simulation names. Each row corresponds to a single simulation and determines where the simulation output and results of the analysis are stored.

In brief, to manually run the whole-cell model: a new bash script, gene list, and experiment list are created on the desktop computer to answer an experimental question. The supercomputer is accessed on the desktop via ftp software, where the new experimental files are uploaded, the planned output folders are created, and MGGRunner.m, runGraphs.m, compareGraphs.m files are confirmed to be present. The supercomputer is then accessed on the desktop via ssh software, where the new bash script is made executable and added to the supercomputer's queuing system to be executed. Once the experiment is complete, the supercomputer is accessed on the desktop via ssh software, where the results of the analysis are moved to /pdf and /fig folders. These folders are accessed on the desktop via ftp software, where the results of the analysis are downloaded. More detailed instructions are contained within the template bash script.

Each wild-type simulation consists of 300 files requiring 0.3 GB. Each gene manipulated simulation can consist of up to 500 files requiring between 0.4 and 0.9 GB. Each simulation takes 5–12 h to complete in real time, 7–13.89 h in simulated time.

**Reporting summary**. Further information on research design is available in the Nature Research Reporting Summary linked to this article.

## Data availability

The databases used to design our in silico experiments, and compare our results to, includes Karr et al.[18] and Glass et al.[27] Supplementary Tables, and Fraser et al. *M. genitalium* G37 genome[19] interpreted by KEGG[38] and UniProt[28] as strain ATCC 33530/NCTC 10195. The initial input and all of the Minesweeper genome simulations completed (raw and transformed output) consist of 4.2 TB of data, and are available from the corresponding author upon reasonable request. All of the GAMA genome simulations transformed output data (ko.db), the binary genome data used to produce Fig. 5, the output .fig files for all simulations referenced in the Supplementary Data (including Minesweeper simulations) are available from our group's Research Data Repository (data-bris) at the University of Bristol, with the identifier[37] https://doi.org/10.5523/bris.1jj0fszzrx9qf2ldcz654qp454. The authors declare that all other data supporting the findings of this study are available within the paper and its supplementary information files.

## Code availability

The code used for this research is openly available on Github (https://github.com/GriersonMarucciLab) This includes the code for Minesweeper and GAMA genome design tools, scripts for statistical analysis, scripts for analysing GO terms, our custom simulation runner, analysis scripts, a template bash script, as well as the bash scripts and text files used to generate the simulations in this paper.

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

## Acknowledgements

We would like to thank the Advanced Computing Research Centre (ACRC) and Bris-SynBio, a BBSRC/EPSRC Synthetic Biology Research Centre, at the University of Bristol for access to the BlueCrystal and Bluegem supercomputers. Special thanks to the HPC and RDSF teams of the ACRC, particularly Dr. Christopher Woods, Simon Burbidge, Matt Williams, and Damian Steer for their help with BlueCrystal, BlueGem, data storage and publication. We would like to thank Jonathan Karr for his advice on running gene knockout simulations using the *M. genitalium* whole-cell model, and for his constructive and enlightening feedback. We would like to thank Anthony Vecchiarelli (Assistant Professor, University of Michigan) and class MCDB 401 ("Building the Synthetic Cell") for conducting a class review of our preprint paper, providing us with constructive and encouraging feedback. We would like to thank John Glass (Professor, JCVI Synthetic Biology and Bioenergy Group) for his constructive and informative feedback. We would like to thank Cameron Matthews and Julia Needham, University of Bristol undergraduates, who conducted simulations to test the inclusion of MG_290 within the phosphonate group as part of summer studentships. L.M. is supported by the Medical Research Council grant MR/N021444/1 to L.M., and by the Engineering and Physical Sciences Research Council grant EP/R041695/1 to L.M. O.C., L.M. and C.G. are supported by a BrisSynBio, a BBSRC/EPSRC Synthetic Biology Research Centre (BB/L01386X/1), flexi-fund grant. O.C. is supported by the Bristol Centre for Complexity Sciences (BCCS) Centre for Doctoral Training (CDT) EP/I013717/1; J.R.G. and S.L. are supported by EPSRC Future Opportunity Scholarships.

## Author contributions

C.G., L.M. and O.C. for attaining initial funding. C.G., L.M., O.P., O.C., J.R.G. and S.L. were involved in ideation. S.L. was involved in analysis and development of Fig. 4. O.C. was responsible for the development and implementation of the *Mycoplasma genitalium* whole-cell model outside of the Covert Lab, Stanford (on Bristol's BlueGem and Blue-Crystal), initial ideation about uses of whole-cell models, GAMA (Methods and Results), development of genome design suite, Fig. 2, Fig. 5, and collaborative theorising on essentiality and minimal genomes. J.R.G. was responsible for the development of automated graphing, Minesweeper (Methods and Results), spreadsheet analysis of in silico results, Table 1, Fig. 1, Fig. 3, collaborative theorising on essentiality and minima, writing of the paper and Supplementary Information, responses and new analysis based on reviewers feedback. C.G., L.M., O.C., O.P. and S.L. were involved in editing and feedback on paper.

## Competing interests

The authors declare no competing interests.
