## [Peer Review File · Nature Communications]

Reviewers' comments:

Reviewer #1 (Remarks to the Author):

The manuscript proposes two computational approaches to generate in silico minimal genomes using the *M. genitalium* whole cell model. The approach proposes minimal genomes that are smaller than the existing minimal cells that have been generated in *M. mycoides*. Given the complexity of generating minimal genomes there is good reason to use computational approaches to engineer minimal genomes that can be tested in silico before generating in the lab.

I have the following comments on the manuscript:

1. I would also like to see greater clarity on how proteins without known function are approached in the study. Publication of JCVI-Syn3.0 suggested that nearly a third of the genes had unknown function. Here, the introduction states that the model " includes the function of every known gene product (401 of the 525 *M. genitalium* genes)". Does this mean that the function for 401 of the 525 genes is known and that only these 401 genes are included in the analysis? If so, it should be clearer how the remaining 124 are treated in the model and considered in the total number of genes that are required.
2. The introduction refers to the recent number of genes in the minimal *M. Mycoides* genome as 413 but my understanding that the Science 2016 paper referred to 473 genes (reference 2)? Which number is correct? If 413, then why is this number correct?
3. In the results section I am further confused the number of genes that are referred to (page 8 line 154) refers to *M. genitalium* cell containing only 256 genes, but this appears to ignore the 124 that from my understanding are considered to be essential as their function is unknown. I think this needs to be clear, how many genes are there in this in silico minimal genome cell?
4. Similarly for GAMA_236, I expect that the other 124 genes have not been included?
5. Figure 3 - some of the y scales for the same property vary between the three cells, it would be easier to compare if they were plotted on the same scale.
6. The results (from line 244) consider the functions that have been removed using Gene Ontology categories. It is difficult to see how a cell would be able to survive based on some of the functions that have been removed e.g. proton transport, DNA recombination and repair, chromosome (segregation and separation). The manuscript makes some comments about these effects but I am left wondering how long such a cell would be able to survive and continue to replicate without these processes present?
7. Line 280 - I would like greater explanation of how it was possible to remove 18 high essentiality genes from GAMA_236. I do not find the reference to the supplementary material helpful
8. The different use of phosphate sources is interesting and the manuscript relates this to the presence of different minimal genomes in different environments, however there is no detail on the environment in which this minimal cell is growing and how the minimal genome might differ in a different environment - this could be a good area for discussion.
9. I feel there is limited detail on the confidence of the in silico minimal genomes that have been generated. The discussion refers to a 3% chance for error (line 353) but I think there needs to be greater consideration of this.
10. I do not follow the sentence that ends online 367 - is it being suggested that the minimal genomes would be generated without the genes of unknown function? This seems unlikely, recent work has proposed important functions for these genes in JCVI-Syn3.0 (Nature Communications 2019 10:3100). It seems naive to think that these can just be removed.
11. I would like to see greater consideration of the limitations to the work. For example the manuscript states that the cells may only function in the first generation of cells - is there a way that they could assess this? This also goes back to my question about the functions that have been removed - has a viable minimal cell really been generated?

Reviewer #2 (Remarks to the Author):

The authors describe their efforts to create reduced-genome versions of a comprehensive cell model of *Mycoplasma genitalium*. They employ two different algorithms, one of which produces a reduction of 145 genes, and the other a reduction of 165 genes. They then perform some further analysis of the resulting simulations. I found this study to be fascinating and an important new chapter in the field of whole-cell modeling and its applications; this was clearly a massive effort in producing the reduced-genome simulations and the authors should be commended for that effort. I also agree with the authors that a future which combines experimental methods with computational strategies such as what they describe will be critical to the future of large-scale cell design. That said, I was hoping for more details in the results and analysis, in particular with respect to the biology, and would love to see such details in a revised manuscript. The best biological part of the story involves phosphate sources - so just to encourage the authors to mention more details of that kind.

* Overall, I would greatly appreciate a detailed narration of the supplementary tables. In its current form, the information is all there but it is very difficult to interpret. I would like to see a detailed description of every table in the supplement, wherein not only is the table described in more detail but the key highlights from the table are listed. For example, early in the main text the authors assert that the results of their initial knockout analysis agrees 97% with the original study. We can look that up in the table, but we also need to hear from them - what were the differences? Is it because of mistakes in the original work, or is something wrong with the new implementation? What are the functions of the genes in question, and does this suggest something about the annotation of those genes? A few paragraphs on this topic would add a lot to the work, and this goes for all of the other tabs in the spreadsheet as well.

* One of the interesting aspects of the work is the several nuances to essentiality, and it would be nice to read more about the biology in the text. For example, in Table 1 and accompanying text, low essentiality genes are described - but the GO terms are not very helpful, and so the annotation should be added to this figure. Also, do either the GAMA or the Minesweeper algorithms predict protective or redundant essentiality? It would seem that *dnaK* is such a case, so then the identity of the cognate gene would be of interest.

* Along those same lines, there's no table and barely any discussion of the 18 high essentiality genes in the main text - surprising given its prominence in the abstract.

* I thought the differences in the simulations - particularly the GAMA-determined model's capacity to produce viable simulations only 18% of the time - was fascinating and warranted further investigation. It seems highly likely that the addition of one gene or a small number of genes to GAMA_236 could bump that percentage back up closer to the Minesweeper model's levels. The drop in RNA production and the overlap between deleted genes for the two models seem like clues that could be followed up on in that regard - something like this: (1) look at the genes that were deleted in GAMA but not Minesweeper, (2) see which of these might be likely to impact RNA production (not necessarily protein production, since it would be downstream), (3) re-add to GAMA and simulate. If the two algorithms could be used in this way to yield a more reduced genome that is more viable, that would be a very exciting result.

* Figure 4 would be much more interesting if the genes were named and highlighted in cases of differences between the models. For example, all genes which were present in GAMA but not Minesweeper could be labeled in orange, and vice versa in blue, on the outside of the circle. This is another example of where the biological detail makes the work more tactile and exciting.

* Now that a reduced-genome *Mycoplasma* exists - even though it's much larger - it seems important to explicitly compare the deletions in that organism to the deletions here. Are there any specific genes that were removable here, but are not in the actual organism?

* Minor: in Figure 5's caption, the last sentence is not proven in the Figure (instead it's in Table 1) and so this should be removed.

Reviewer Comments and Rebuttal (NCOMMS-19-20829)

Reviewer 1

1. I would also like to see greater clarity on how proteins without known function are approached in the study. Publication of JCVI-Syn3.0 suggested that nearly a third of the genes had unknown function. Here, the introduction states that the model “ includes the function of every known gene product (401 of the 525 *M.genitalium* genes)”. Does this mean that the function for 401 Of the 525 genes is known and that only these 401 genes are included in the analysis? If so, it should be clearer how the remaining 124 are treated in the model and considered in the total number of genes that are required.

Agreed. This needed clarifying (linked to other comments by Reviewer 1: 3, 4, 10).

The section now reads: Line 61 “It is the only existing model of a cell’s individual molecules that includes the function of every known gene product (401 of the 525 *M.genitalium* genes) making it capable of modelling genes in their genomic context 18. 124 genes of unknown function are not modelled in-silico, but as some unknown function genes have proposed fundamental functions 20, these are assumed to be essential in-vivo (so are added in our in-vivo predictions).”

Reference 20 is the suggested reference from Comment 10 (Nature Communications 2019 10:3100)

This has been adopted throughout the paper e.g.

Line 145 “In total, Minesweeper deleted 145 genes, creating an in-silico *M.genitalium* cell containing 256 genes (named Minesweeper_256) and predicting an in-vivo minimal genome of 380 genes.”

2. The introduction refers to the recent number of genes in the minimal *M. Mycoides* genome as 413 but my understanding that the Science 2016 paper referred to 473 genes (reference 2)? Which number is correct? If 413, then why is this number correct?

Agreed. Apologies to the reviewers, this sentence incorrectly hosted multiple references when it should have only referenced Glass et al 2017 (Figure 1, Minimal Cells-Real and Imagined, Cold Spring Harb. Perspect. Biol., 2017), where the 413 gene prediction comes from. This has been corrected (Line 74).

3. In the results section I am further confused the number of genes that are referred to (page 8 line 154) refers to *M.genitalium* cell containing only 256 genes, but this appears to ignore the 124 that from my understanding are considered to be essential as their function is unknown. I think this needs to be clear, how many genes are there in this in silico minimal genome cell?

Agreed. This has been changed and hopefully, with other clarifications in the paper, it is now clearer. It now reads: Line 145 “In total, Minesweeper deleted 145 genes, creating an in-silico *M.genitalium* cell containing 256 genes (named Minesweeper_256) and predicting an in-vivo minimal genome of 380 genes.”

4. Similarly for GAMA_236, I expect that the other 124 genes have not been included?

Agreed. Similarly to Comment 3, this now reads: Line 177 “In total, the smallest GAMA-reduced in-silico genome deleted 165 genes, creating an in-silico M.genitalium genome of 236 genes (named GAMA_236), and predicting an in-vivo minimal genome of 360 genes.”

5. Figure 3 - some of the y scales for the same property vary between the three cells, it would be easier to compare if they were plotted on the same scale.

Agreed. Have replotted Figure 3 (pg 30) so that each property has the same y scales.

6. The results (from line 244) consider the functions that have been removed using Gene Ontology categories. It is difficult to see how a cell would be able to survive based on some of the functions that have been removed e.g. proton transport, DNA recombination and repair, chromosome (segregation and separation). The manuscript makes some comments about these effects but I am left wondering how long such a cell would be able to survive and continue to replicate without these processes present?

Agreed. Further research has been done to clarify the removal of these functions (line 259-280), highlighting the still present ATP production pathways (associated with proton transport), assessing the state of DNA repair in the in-silico cells, and clarifying the continued presence of chromosome segregation in the modelled cells. Consideration of the cell survival is included as part of a larger discussion (line 406-431).

7. Line 280 - I would like greater explanation of how it was possible to remove 18 high essentiality genes from GAMA_236. I do not find the reference to the supplementary material helpful

Agreed (a similar point was raised by the other reviewer). This has been further investigated and addressed (see lines 368-390, 406-431, Table 2), and the reference to the high essential genes being removed from the abstract. We theorise that the high essential genes were able to be removed only in cases of high initial conditions, highlighting issues with single generation models.

8. The different use of phosphate sources is interesting and the manuscript relates this to the presence of different minimal genomes in different environments, however there is no detail on the environment in which this minimal cell is growing and how the minimal genome might differ in a different environment - this could be a good area for discussion.

Agreed. Have added a section to Methods (Line 474-480) detailing the environment the minimal cell is growing, referenced on Line 341, where how it might differ in a different environment is now briefly discussed.

9. I feel there is limited detail on the confidence of the in silico minimal genomes that have been generated. The discussion refers to a 3% chance for error (line 353) but I think there needs to be greater consideration of this.

Agreed. This specific point has been replaced with a larger and more general discussion (see line 406-431). The ultimate test of our predictions will be to test our minimal genome predictions in vivo, which is beyond the scope of this paper.

10. I do not follow the sentence that ends on line 367 - is it being suggested that the minimal genomes would be generated without the genes of unknown function? This seems unlikely, recent work has proposed important functions for these genes in JCVI-Syn3.0 (Nature Communications 2019 10:3100). It seems naive to think that these can just be removed.

Agreed. Have removed the comment, and rewritten as: Line 426 “We believe that our in-silico shared deletions could predict a viable *Mycoplasma genitalium* minimal cell. Given that the impact of the unmodelled genes is unknown (e.g. if they perform a unique essential function with a gene / gene product that has already been removed the in-vivo cell will not survive), until these predictions are tested experimentally we cannot firmly state how long our predicted reduced in-vivo cells would survive and replicate, and whether they represent a truly minimal *M.genitalium* genome. ”

11. I would like to see greater consideration of the limitations to the work. For example the manuscript states that the cells may only function in the first generation of cells - is there a way that they could assess this? This also goes back to my question about the functions that have been removed - has a viable minimal cell really been generated?

Agreed. Unfortunately, there is not a way to assess this in-silico, and it is not possible to assess this in-vivo without further funding and time. This has been discussed generally (see line 406-431).

Reviewer 2

General: More details in the results and analysis, in particular with respect to the biology.

1. Detailed narration of the supplementary tables. In its current form, it is very difficult to interpret. I would like to see a detailed description of every table in the supplement (table described in more detail and key highlights listed).

For example, early in the main text the authors assert that the results of their initial knockout analysis agrees 97% with the original study. We can look that up in the table, but we also need to hear from them - what were the differences? Is it because of mistakes in the original work, or is something wrong with the new implementation? What are the functions of the genes in question, and does this suggest something about the annotation of those genes? A few paragraphs on this topic would add a lot to the work, and this goes for all of the other tabs in the spreadsheet as well.

Agreed. Each tab now has a descriptive paragraph, with the key results highlighted, and the tables have been reformatted and labelled. To answer the specific question highlighted (taken from Tab C): From reading Karr et al's Supplementary Table and investigating the Whole Cell model code base, we believe that the classifications were based on 5x simulations per gene, the results were then averaged together, and the classifications assigned computationally to averaged data. We completed 10x simulations per gene, and assigned classifications manually. The differences may be due to edge cases that were missed by averaging and computational assessment.

2. One of the interesting aspects of the work is the several nuances to essentiality, and it would be nice to read more about the biology in the text. For example, in Table 1 and accompanying text, low essentiality genes are described - but the GO terms are not very helpful, and so the annotation should be added to this figure.

Agreed. In the table of low essentiality genes (pg 33) the GO terms have been replaced with UniProt annotations, which tend to be more informative.

Also, do either the GAMA or the Minesweeper algorithms predict protective or redundant essentiality? It would seem that dnaK is such a case, so then the identity of the cognate gene would be of interest.

Agreed. The large number of simulated knockouts produced by GAMA and Minesweeper can be used to uncover protective / redundant essentiality by cross-comparison, but the algorithms do not themselves predict it. However, dnaK's cognate gene has now been explored. See Line 351-358.

3. Along those same lines, there's no table and barely any discussion of the 18 high essentiality genes in the main text - surprising given its prominence in the abstract.

Agreed (a similar point was raised by the other reviewer). The high essentiality table has been moved from the SI to the main text (Table 2, pg 34), and updated with UniProt annotations. This has been further investigated and addressed (see lines 368-390, 406-431, and Table 2), and the reference to the high essential genes being removed from the abstract.

4. I thought the differences in the simulations - particularly the GAMA-determined model's capacity to produce viable simulations only 18% of the time - was fascinating and warranted further investigation. It seems highly likely that the addition of one gene or a small number of genes to GAMA_236 could bump that percentage back up closer to the Minesweeper model's levels. The drop in RNA production and the overlap between deleted genes for the two models seem like clues that could be followed up on in that regard - something like this: (1) look at the genes that were deleted in GAMA but not Minesweeper, (2) see which of these might be likely to impact RNA production (not necessarily protein production, since it would be downstream), (3) re-add to GAMA and simulate. If the two algorithms could be used in this way to yield a more reduced genome that is more viable, that would be a very exciting result.

We are grateful for this excellent suggestion which has helped us to improve our paper. We followed this line of investigation: conducting independent single knock-ins of the 18 high essential genes to GAMA_236 (100 reps), conducting simulations of a powerset (123 combinations) of 7 high essential genes related to RNA production (10 reps), with the top four from the powerset (that produced the most dividing cells) repeated (100 reps). These results are available in the Supplementary Table O and P. The independent single knock-ins (18* GAMA_237) produced division rates between 13% and 33% (-5% to +15% compared to GAMA_236). The RNA production powerset knock-ins produced division rates between 14% and 23% (-4% to +5%). The highest division rate was 33%, due to a single independent gene knock-in of MG_270. This was graphed and compared to figure 3. The paper has been updated throughout, to consider GAMA_237 in light of these results (specifically discussed line 195-202). Figure 3 has been updated (page 30), and Supplementary Figure 1 has been created which compares the genomes from the new Figure 3 with GAMA_236.

5. Figure 4 would be much more interesting if the genes were named and highlighted in cases of differences between the models. For example, all genes which were present in GAMA but not Minesweeper could be labeled in orange, and vice versa in blue, on the outside of the circle. This is another example of where the biological detail makes the work more tactile and exciting.

Agreed. These changes have been made to Figure 4 (pg 31).

6. Now that a reduced-genome Mycoplasma exists - even though it's much larger - it seems important to explicitly compare the deletions in that organism to the deletions here. Are there any specific genes that were removable here, but are not in the actual organism?

We have completed the comparison as suggested by the reviewer. Firstly, we matched Mycoplasma genitalium and JCVI-Syn3.0 genes using BLAST (tblastn), matching only 264/473 JCVI-Syn3.0 genes (56%). Of the 141 shared deletions, 73 had no BLAST match with JCVI-Syn3.0. There were 15 shared deletions and 53 deletions that were removed in-silico, but not in JCVI-Syn3.0. We conclude that an explicit comparison between the reduced Mycoplasma genomes is difficult and inconclusive, mainly due to the differences in the considered species (JCVI-Syn3.0 is a reduction of JCVI-Syn1.0, which is based on Mycoplasma mycoides). The new data is displayed in Supplementary Table R, and referenced as part of the discussion on Line 418-423.

7. Minor: in Figure 5's caption, the last sentence is not proven in the Figure (instead it's in Table 1) and so this should be removed.

Agreed. Have removed this sentence from Figure 5's caption (pg 32).

REVIEWERS' COMMENTS:

Reviewer #1 (Remarks to the Author):

My comments have been extensively addressed by the authors. I think this is a significant piece of work that is of considerable interest to the minimal genome research field

Reviewer #2 (Remarks to the Author):

The authors present a revised manuscript in which they performed significant subsequent analysis to resolve both my and the other reviewer's concerns. In my opinion, this has led to a greatly improved study. I particularly appreciated their efforts to bring more biological discussion to the text, this made a big difference in the readers' ability to appreciate what has been done, and also led to a significant corrective addition with regard to the high essential genes and the failure of the model to predict multiple generation behaviors. I also thought that the enhanced treatment of the GAMA 236 strain, in particular to produce GAMA 237, was interesting and brought the methods together in a novel way. I support publication and congratulate the authors on their exciting work.

Referee Responses NCOMMS-19-20829A

Reviewer #1

My comments have been extensively addressed by the authors. I think this is a significant piece of work that is of considerable interest to the minimal genome research field

Reviewer #2

The authors present a revised manuscript in which they performed significant subsequent analysis to resolve both my and the other reviewer's concerns. In my opinion, this has led to a greatly improved study. I particularly appreciated their efforts to bring more biological discussion to the text, this made a big difference in the readers' ability to appreciate what has been done, and also led to a significant corrective addition with regard to the high essential genes and the failure of the model to predict multiple generation behaviors. I also thought that the enhanced treatment of the GAMA 236 strain, in particular, to produce GAMA 237, was interesting and brought the methods together in a novel way. I support publication and congratulate the authors on their exciting work.

- ***We would like to thank the reviewers for their constructive and positive critiques of the first submission, which have made for a better paper, and their kind comments relating to the second submission.***